# Identification of Cognitive Strategies Used by Cancer Patients as a Basis for Psychological Self-Support during Oncological Therapy

**DOI:** 10.3390/ijerph19159243

**Published:** 2022-07-28

**Authors:** Karolina Osowiecka, Anna Kieszkowska-Grudny, Radosław Środa, Dominik Olejniczak, Monika Rucińska

**Affiliations:** 1Department of Psychology and Sociology of Health and Public Health, School of Public Health, University of Warmia and Mazury in Olsztyn, Warszawska 30, 10-082 Olsztyn, Poland; 2Minds of Hope, Sokołowska 9, 01-142 Warsaw, Poland; info@mindsofhope.eu; 3Instytut Bez Stresu, Zamenhofa 5, 00-165 Warsaw, Poland; 4Department of Neurosurgery, Military Institute of Medicine, Szaserów 128, 04-141 Warsaw, Poland; sr.radoslaw@gmail.com; 5Department of Public Health, Faculty of Health Science, Medical University of Warsaw, Ul. Nielubowicza 5, 02-097 Warsaw, Poland; dolejniczak@wum.edu.pl; 6Department of Oncology, Collegium Medicum, University of Warmia and Mazury in Olsztyn, Ul. Wojska Polskiego 37, 10-228 Olsztyn, Poland; m_rucinska@poczta.onet.pl

**Keywords:** cancer, cognitive strategies, psychological support

## Abstract

Background: Cancer diagnosis is associated not only with health problems but also with psycho-social disability. Both medical and non-medical problems have impacts on cancer patients’ quality of life. The aim of the study was the identification of cognitive emotion regulation strategies among cancer patients during radiotherapy. Methods: The study was conducted on 78 radically treated cancer patients (median 63 years). A Cognitive Emotion Regulation Questionnaire (CERQ) was used. Results: Cancer patients mostly used acceptance, positive refocusing, putting into perspective and refocus on planning. Age was inversely correlated with refocus on planning. Patients with higher levels of education tended to use rumination and catastrophizing less frequently (*p* < 0.05). Adaptive cognitive strategies based on putting into perspective were more frequently used by professionally active patients (*p* < 0.05). Patients who lived in cities used positive refocusing and putting into perspective significantly often and patients who lived in villages more frequently used catastrophizing (*p* < 0.05). Among lung cancer patients, catastrophizing and rumination were popular (*p* < 0.05) and breast cancer patients rarely used non-adaptive cognitive strategies. Conclusion: Cancer patients tended to use adaptive cognitive strategies. Personalized psychological support should be focused on lung cancer patients and older, less educated, unemployed individuals and people who lived in the countryside.

## 1. Introduction

Cancer diagnosis is a difficult emotional situation for patients. Patients face severe illness with poor prognosis. New challenges connected with diagnostic procedures, treatment and health care organization appear. Cancer patients have a lot of problems, not only related to their health but also non-medical, such as: organizational, social, spiritual, psycho-emotional and communicational [1]. Cancer diagnosis is related with temporary or permanent disability on physical, psychological and social levels [2]. Patients’ daily life changes in various areas: reduced ability to work and activities in free time, influence on social role, relations with relatives and friends and decreased self-confidence. Therefore, the hierarchy of cancer patient’s needs is changed. Patients respond in some way to the distress of the disease, switching between behavioral, emotional and cognitive strategies. Emotional problems among cancer patients were strongly associated with lower quality of life and higher symptom burden [3]. Coping with distress, patients use different cognitive emotion regulation strategies. The term “cognitive strategies” refers to using the mind (cognition) to manage the intake of emotionally arousing information [4]. The cognitive processes may help to regulate emotions during or after stressful experiences. Some cognitive strategies are adaptive and beneficial, for example, acceptance (thoughts of acceptance of the experienced event), positive refocusing (thinking about pleasant things instead of a negative event), refocusing on planning (thinking about how to deal with a negative event) and positive reappraisal (attributing a positive meaning to a negative event). The meta-analysis review [5] showed that acceptance seems to have a crucial importance in psychological adjustment to the illness and in reducing patients’ distress. Acceptance, planning, positive refocusing and positive reappraisal as coping strategies were negatively related to anxiety and depression [4,5,6,7]. Li et al. [8] reported that acceptance and positive reappraisal had positive effects on quality of life among women with breast cancer undergoing treatment. In contrast, some strategies could have maladaptive and destructive impacts, such as resignation (belief that there is nothing to do to change or control the situation), rumination (thinking about the feelings associated with a negative event) and catastrophizing (emphasizing extremely negative thoughts associated with an experience). Resignation, catastrophizing and rumination were associated with higher anxiety and depressive symptoms and caused lower quality of life [4,5,8]. Consumption of nicotine and alcohol seem to be destructive strategies for coping with stress among patients [9]. Religious coping attitudes could be divided into positive and negative. Positive religious coping style could influence better compliance to the cancer diagnosis and treatment [10,11]. Among cancer patients undergoing chemotherapy, positive religious coping increased the level of hope [12]. In contrast, negative religious coping, such as the belief that illness is a punishment from God, could influence feelings of hopelessness [13]. Adaptive strategies may help patients to face cancer. Recognizing strategies used by cancer patients to cope with distress is a crucial issue to proper psychological support during cancer treatment and cooperation with oncologists. Cancer diagnosis and therapy have impacts on quality of life [14]. Quality of life among cancer patients was positively correlated with lower symptom burden and better spiritual well-being [15,16,17,18]. Depression has negative impacts on quality of life [19,20]. In Poland, the problem of psychological and social cancer-related disability of cancer patients is still unresolved. There is a lack of studies related to cognitive strategies adopted by patients.

The aim of the study was the identification of cognitive emotion regulation strategies among cancer patients during radiotherapy.

## 2. Materials and Methods

The study was conducted on a group of 78 patients who were treated with radical radiotherapy for malignant neoplasm between April 2018 and October 2018 in oncological centers in Poland (The Military Institute of Medicine in Warsaw; the Maria Sklodowska-Curie Institute—Oncology Center in Warsaw; and the Hospital of the Ministry of Internal Affairs with the Warmia and Mazury Oncology Center in Olsztyn).

Patients: The inclusion criteria were: cancer diagnosis, age ≥ 18 years old, actively being treated for cancer and a signed consent to participate in the study.

Questionnaire: A questionnaire used for the study was a standardized Cognitive Emotion Regulation Questionnaire (CERQ) [4]. The original version of the CERQ was conducted in Dutch and it has been validated in adolescents and adults [4,21]. The questionnaire was adapted to the Polish population by Marszal-Wisnieska and Fijalkowska-Stanik [22]. The CERQ was created to estimate the conscious cognitive components of emotion regulation related to stressful moments in life. It assesses five adaptive strategies (acceptance, refocus on planning, putting into perspective, positive refocusing and positive reappraisal) and four non-adaptive strategies (self-blame, rumination, catastrophizing and blaming others). The internal consistency reliability was estimated using Cronbach’s α coefficient. Acceptance had acceptable internal reliability (0.52) and the other subscales had good internal reliability (≥70; putting into perspective, positive refocusing, self-blame, rumination, catastrophizing) and very good internal reliability (≥80; refocus on planning, positive reappraisal, blaming others) [22]. The full version of the CERQ consisted of 36 questions with a Likert scale from 1 (never/almost never) to 5 (always/almost always). “Acceptance” relates to thinking that it should be accepted because it happened. “Positive refocusing” relates to having positive and pleasant thoughts instead of thinking about stressful events. “Refocus on planning” relates to thinking about what and how to do the best in this situation. “Positive reappraisal” relates to giving a positive meaning to the negative events in terms of personal growth. “Putting into perspective” relates to relativizing the negative situation compared to other worse situations or the ability to look at stressful situations from the perspective of time. “Self-blame” relates to blaming only oneself for bad experiences. “Rumination” relates to often thinking and overthinking about the feelings and thoughts related to negative events. “Catastrophizing” relates to constantly emphasizing extremely negative thoughts and scenarios associated with a bad experience. “Blaming others” relates to blaming others for bad events that one has experienced oneself. In the recent study, a shortened version of the CERQ was used; therefore, positive reappraisal strategy was not included.

Demographic data (gender, age, education, place of residence, marital status and professional activity) were collected using an additional questionnaire prepared specially for this study. Patients were individually interviewed by a psycho-oncologist or psychologist.

The study protocol was approved by the Ethics Committee of the University of Warmia and Mazury in Olsztyn (no 2/2018). Participation in the study was voluntary. All study participants gave their consent and signed it.

### Statistical Analysis

The characterization of the study group was conducted using descriptive statistics. Medians with 25–75% interquartile ranges (IQR) of various cognitive strategies were estimated. The distribution of variables was compared with the theoretical normal distribution using the Shapiro–Wilk test. The differences between the subgroups were analyzed with either the Mann–Whitney (for 2 subgroups) or the Kruskal–Wallis test (for more than 2 subgroups). Linear correlation was tested by Spearman’s rank correlation coefficient. A *p*-value of <0.05 was considered to be significant. The analysis was conducted using Statistica software, version 13 (http://statistica.io accessed on 20 February 2022), TIBCO Software Inc., Krakow, Poland (2017)).

## 3. Results

### 3.1. Characteristics of the Group

The study was carried out on a group of 78 cancer patients. There were 41 women (53%) and 37 men (47%) aged 24–85 years (median age 63 years). More than half of patients had secondary levels of education (56%). A total of 46% of analyzed patients were pensioners and 44% of patients were professionally active. Most of the respondents were married (76%), and lived in cities (82%). Breast cancer, head and neck cancer and lung cancer were the most frequent among analyzed patients (Table 1).

### 3.2. Cognitive Strategies

Cancer patients tended to use adaptive cognitive strategies, such as: acceptance (median 4.25; 25–75% IQR 3.75–4.75), positive refocusing (median 4.25; 25–75% IQR 3.25–4.75), putting into perspective (median 4.0; 25–75% IQR 3.25–4.5) and refocus on planning (median 3.75; 25–75% IQR 3.0–4.5). Other patients developed non-adaptive cognitive strategies, including self-blame (median 2.5; 25–75% IQR 1.5–3.3), rumination (median 2.5; 25–75% IQR 1.5–3.25), catastrophizing (median 2.25; 25–75% IQR 1.5–3.25) and blaming others (median 1.0; 25–75% IQR 1.0–1.5) (Figure 1).

### 3.3. Cognitive Strategies Due to Various Factors

There were no significant differences in any of the analyzed cognitive emotion strategies due to sex and marital status (*p* > 0.05). Age was inversely correlated with one of the adaptive cognitive strategies—refocus on planning (*p* < 0.05; r = −0.25) There were no associations between age and all other cognitive strategies. A significant correlation was noted between education level of patients and non-adaptive cognitive strategies: rumination (*p* = 0.002) and catastrophizing (*p* = 0.009). Patients with higher levels of education tended to use rumination (*p* = 0.002) and catastrophizing (*p* = 0.02) less frequently in comparison with patients who graduated from primary or secondary school (*p* = 0.03). In the case of adaptative cognitive strategies, there were no significant differences. Adaptive cognitive strategies based on putting into perspective were more frequently used by professionally active patients in comparison with unemployed patients (*p* = 0.045). There were no associations between professional activity and any of the other cognitive strategies. Patients who lived in cities used adaptive cognitive strategies significantly often: positive refocusing (*p* = 0.04) and putting into perspective (*p* = 0.02). In the case of non-adaptive cognitive strategies, a significant difference was observed only in catastrophizing. Patients who lived in villages more frequently used catastrophizing in comparison with respondents who lived in cities (*p* = 0.004). Cancer localization had an impact on non-adaptive emotion regulation strategies. In the analysis, the three most frequent cancers were included (breast cancer, head and neck cancer and lung cancer). Lung cancer patients most frequently used catastrophizing (*p* = 0.03) and rumination (*p* = 0.01). Breast cancer patients used non-adaptive cognitive strategies least often (Table 2; Figure 2).

### 3.4. The Relations between Different Cognitive Strategies Were Estimated

Regarding acceptance, it showed a relation directly proportional to positive refocusing (r = 0.415; *p* < 0.001) and putting into perspective (r = 0.506; *p* < 0.001). There were no statistically significant correlations between acceptance and focus on planning or any of the non-adaptive cognitive strategies. Positive refocusing presented a directly proportional relationship with putting into perspective (r = 0.383; *p* = 0.001) and refocus on planning (r = 0.407; *p* < 0.001). The inverse relation was observed between positive refocusing and rumination (r = −0.244; *p* = 0.033) and catastrophizing (r = −0.257; *p* = 0.025). There were no statistically significant correlations between putting into perspective or any of the other cognitive strategies. Regarding refocus on planning, it showed a directly proportional correlation to blaming others (r = 0.269; *p* = 0.019) and it was the only one positive correlation between adaptive and non-adaptive cognitive strategies. In the case of non-adaptive strategies, there were directly proportional correlations between self-blame and rumination (r = 0.406; *p* < 0.001), self-blame and catastrophizing (r = 0.333; *p* = 0.003), rumination and catastrophizing (r = 0.759; *p* < 0.001), rumination and blaming others (r = 0.311; *p* = 0.006) and catastrophizing and blaming others (r = 0.309; *p* = 0.007). The correlation between self-blame and blaming others was not noted (Table 3).

## 4. Discussion

Emotional distress is a typical issue faced by cancer patients. One third of cancer patients experienced distress. However, the level of emotional distress is individual. Patients diagnosed with lung, gynecological, breast and gastrointestinal cancer more often suffered with emotional problem-related distress in comparison with prostate and hematological cancer patients [23]. The level of emotional distress is the highest in the case of lung cancer patients and it was shown by 43% of patients [24]. Younger age and lower incomes are associated with greater emotional distress faced by patients [3,23]. Morrison et al. [3] reported that women, the employed, current smokers and patients in more advanced lung cancer stages experienced emotional problems at a significantly higher rate. Individual tailored strategies for coping with cancer-related problems and negative mood are very important for cancer patients. Bergerot et al. [23] showed a positive correlation between level of distress and cancer patients’ need of psychological support. Therefore, emotional distress screening is really important to identify unmet patients’ needs.

Cognitive emotion regulation strategies help people to keep control over their emotions during or after the experience of threatening or stressful events [4,21]. Some strategies reduce stress—adaptive, whereas some increase it—non-adaptive. Adaptive strategies include positive reappraisal, positive refocusing, acceptance and refocusing on planning. Non-adaptive strategies include self-blame, blaming others, rumination and catastrophizing. Adaptive strategies have been shown to be essential for initiating, motivating and organizing adaptive behavior in cancer patients [25]. Non-adaptive strategies could lead to maladaptive behaviors [25]. Kvillemo et al. [26], in a meta-analysis of 78 studies on breast cancer patients, showed that the use of adaptive cognitive strategies was associated both with better well-being and physical health. In our study, cancer patients mostly used adaptive cognitive strategies, especially acceptance and positive refocusing. Patients almost never used blaming others. However, Wang et al. [27] found that gynecological cancer patients largely use self-blame and blaming others, and little use acceptance and putting into perspective. In our study, there were only 8 gynecological cancer patients, but also among them the most popular strategies were acceptance as an adaptive strategy and catastrophizing as non-adaptive strategy.

There were no significant correlations between cognitive strategies and sex and marital status. Some non-adaptive strategies were chosen by single people in Calderon et al.’s analysis [28]. Santos et al. presented that single people experienced a more fatalistic strategy [29]. More support from friends and family has an effect on using more effective coping strategies among cancer patients [30,31,32]. In our study, older patients used refocus on planning less frequently in comparison with younger patients. In other strategies, there were no significant correlations with age. Calderon et al. [28] noted that younger age was associated with the use of adaptive coping strategies. Other authors showed that younger breast cancer patients used more efficacious coping strategies [33,34]. In the current study, respondents with low educational levels more frequent used non-adaptive strategies, such as rumination and catastrophizing. Similar results were presented in Calderon et al.’s analysis [28]. Other authors also noted that low educational level was related to more helplessness and cognitive avoidance [29]. Professional activity was positively related to using a putting into perspective strategy. People who lived in cities used putting into perspective and positive refocusing more often. However, people who lived in villages used non-adaptive strategies more often, mainly catastrophizing. Due to cancer localization, lung cancer patients used catastrophizing and rumination more often in comparison with head and neck cancer and breast cancer. Women with breast cancer used non-adaptive strategies less often.

Anxiety was not evaluated in the current study. Da Silva et al. [35] noticed that patients with low levels of anxiety were more likely to focus on problem solving but not on emotions, in contrast to patients with higher levels of anxiety, although it is worth noting that it is not entirely clear whether the low level of anxiety was due to the focus on problem solving or whether focus on problem solving was possible due to the lower level of anxiety. Garnefski et al. [4] also showed that people who adopt adaptive strategies report fewer depression and anxiety symptoms than people who use less adaptive strategies. Anxiety and depression were strongly proportionally correlated with rumination and self-blame. A converse correlation was noted, especially with positive refocusing [4]. In our study, one of the two most-used strategies was positive refocusing. However, breast cancer patients, who have low levels of anxiety tend to use problem-focused coping strategies more often, which may have a positive impact on adaptation to breast cancer [35,36]. According to other authors, strategies focused on emotion are related to avoiding the problem and it could cause difficulties in adjusting patients to the new reality in contrast to patients who are focused on the problem [37,38].

### Limitations and Future Perspectives

There were some limitations in the study. The sample size was relatively small. All patients were under radiotherapy and had had some other oncological treatment before. It was not analyzed, but chemotherapy, hormone therapy and surgery could have had some impact on their cognitive strategies. In the study, only one questionnaire was used. It will be interesting to evaluate the results obtained by the CERQ using other tools. The cognitive strategies used by cancer patients might correlate with personality and depression/anxiety/psychological distress variables. Therefore, some personality questionnaires and mental health distress markers should be applied in future studies.

## 5. Conclusions

It is important to recognize strategies applied by cancer patients to cope with emotions related to cancer diagnosis. Fortunately, most cancer patients tended to use adaptive cognitive strategies. Personalized psychological support should be focused on patients who choose non-adaptive cognitive strategies: people who are lung cancer patients, elders, less educated, unemployed or live in the countryside. Employing adaptive cognitive strategies could help cancer patients to deal with disease-related emotions and improve quality of life.

## Figures and Tables

**Figure 1 ijerph-19-09243-f001:**
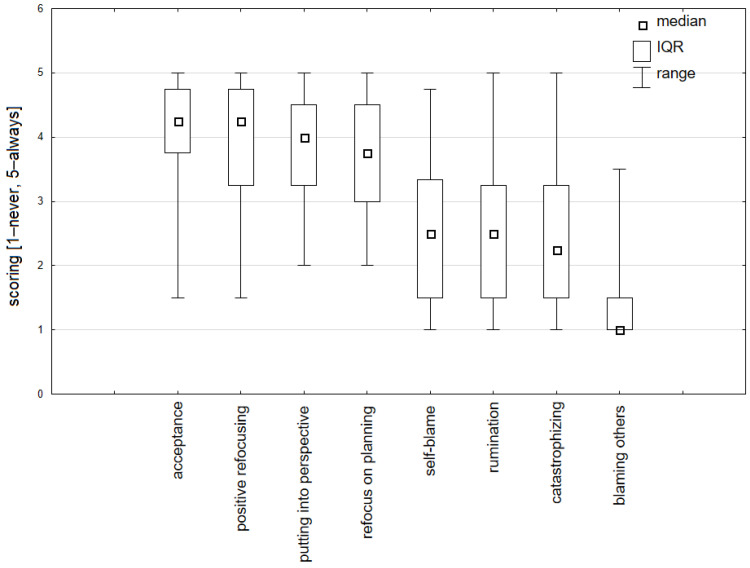
Cognitive strategies used by cancer patients during radical therapy.

**Figure 2 ijerph-19-09243-f002:**
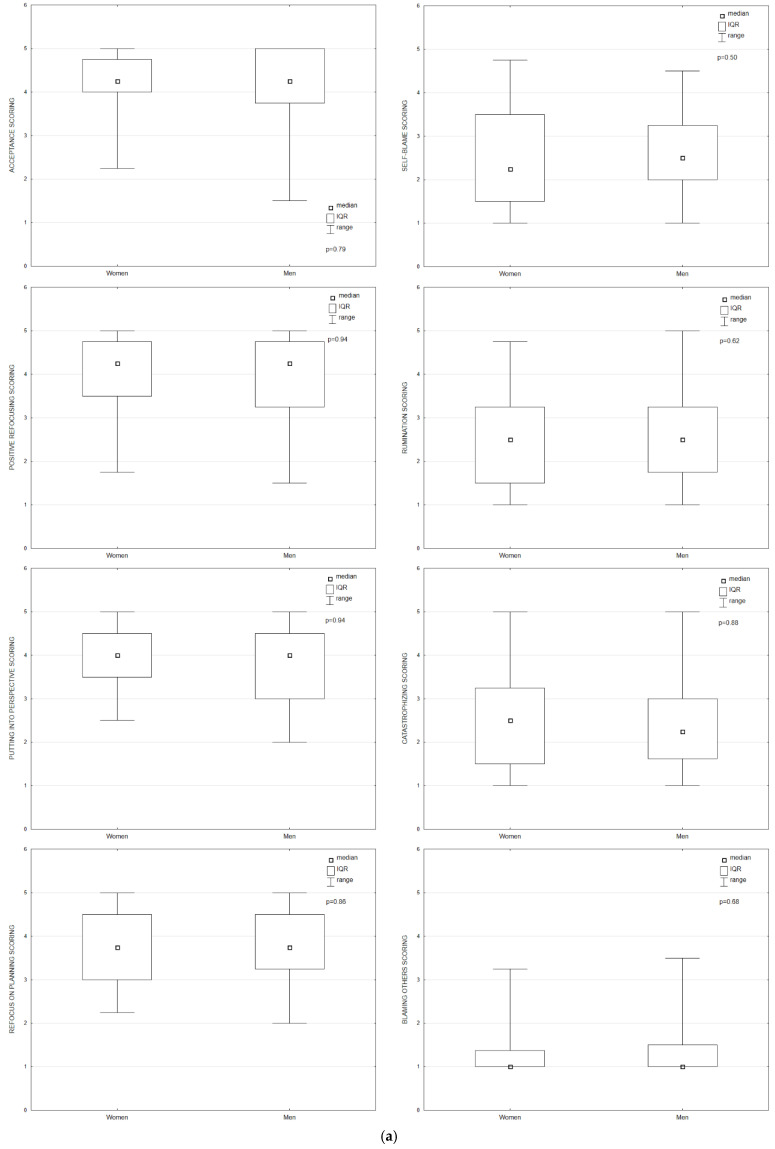
Cognitive strategies’ scoring (1—never, 5—always) due to various factors: (**a**) sex; (**b**) age, r—correlation coefficient; (**c**) education; (**d**) professional activity; (**e**) marital status; (**f**) place of residence; (**g**) cancer localization.

**Table 1 ijerph-19-09243-t001:** Characteristic of patients.

		n = 78	(%)
Age median (IQR)		63 (53–69)	
Gender			
	female	41	(52.6)
	male	37	(47.4)
Education			
	primary	13	(16.7)
	secondary	44	(56.4)
	higher	21	(26.9)
Professional activity			
	student	1	(1.3)
	employed	34	(43.6)
	unemployed	7	(9.0)
	pensioner	36	(46.1)
Marital status			
	married	59	(75.6)
	single	12	(15.4)
	widow/widower	7	(9.0)
Place of residence			
	city	64	(82.1)
	village	14	(17.9)
Cancer			
	breast	18	(23.1)
	head and neck	16	(20.5)
	lung	15	(19.2)
	gynecological	8	(10.3)
	prostate	7	(9.0)
	esophagus and stomach	4	(5.1)
	rectum	3	(3.8)
	others	7	(9.0)

IQR—interquartile range.

**Table 2 ijerph-19-09243-t002:** Cognitive strategies among cancer patients due to various factors.

Variables		Acceptance	*p*-Value	Positive Refocusing	*p*-Value	Putting into Perspective	*p*-Value	Refocus on Planning	*p*-Value	Self-Blame	*p*-Value	Rumination	*p*-Value	Catastrophizing	*p*-Value	Blaming Others	*p*-Value
Sex	median (25–75% IQR)																
	women	4.25 (4.0–4.75)	0.79	4.25 (3.5–4.75)	0.94	4.0 (3.5–4.5)	0.94	3.75 (3.0–4.5)	0.86	2.25 (1.5–3.5)	0.50	2.5 (1.5–3.25)	0.62	2.5 (1.5–3.25)	0.88	1.0 (1.0–1.38)	0.68
	men	4.25 (3.75–5.0)		4.25 (3.25–4.75)		4.0 (3.0–4.5)		3.75 (3.25–4.5)		2.5 (2.0–3.25)		2.5 (1.75–3.25)		2.25 (1.63–3.0)		1.0 (1.0–1.5)	
Age	r	r = 0.16	>0.05	r = 0.001	>0.05	r = 0.06	>0.05	r = −0.25	<0.05	r = 0.009	>0.05	r = −0.1	>0.05	r = −0.04	>0.05	r = −0.04	>0.05
Education	median (25–75% IQR)																
	primary	4.5 (4.0–4.75)	0.46	4.0 (3.0–4.5)	0.47	4.0 (3.0–4.25)	0.89	3.5 (2.75–4.5)	0.16	2.5 (2.0–3.0)	0.30	3.25 (2.5–4.0)	0.002	3.38 (1.38–4.5)	0.009	1.25 (1.0–1.75)	0.05
	secondary	4.0 (3.75–4.75)		4.25 (3.38–4.75)		4.0 (3.25–4.75)		3.75 (3.5–4.6)		2.5 (1.8–3.5)		2.5 (1.9–3.3)		2.6 (2.0–3.1)		1.0 (1.0–1.75)	
	higher	4.5 (4.0–5.0)		4.25 (3.5–5.0)		4.0 (3.5–4.5)		3.5 (3.0–4.0)		2.0 (1.5–3.3)		1.5 (1.25–2.5)		2.0 (1.0–2.25)		1.0 (1.0–1.13)	
Professional activity	median (25–75% IQR)																
	employed	4.25 (3.75–5.0)	0.48	4.25 (4.0–4.75)	0.17	4.0 (3.5–4.75)	0.049	4.0 (3.0–4.5)	0.38	2.25 (1.5–3.5)	0.54	2.5 (1.75–3.25)	0.95	2.5 (1.75–2.75)	0.49	1.0 (1.0–1.5)	0.24
	unemployed	4.0 (2.75–4.5)		3.5 (2.75–4.75)		3.0 (2.5–4.0)		3.5 (3.25–4.25)		2.5 (2.5–3.25)		2.0 (2.0–4.25)		3.0 (1.5–4.25)		1.5 (1.0–2.25)	
	pensioner	4.13 (4.0–4.9)		4.0 (3.0–4.5)		4.0 (3.13–4.5)		3.75 (3.0–4.13)		2.5 (1.88–3.42)		2.7 (1.5–3.4)		2.25 (1.63–3.38)		1.0 (1.0–1.5)	
Marital status	median (25–75% IQR)																
	married	4.25 (4.0–5.0)	0.66	4.25 (3.0–4.75)	0.78	4.0 (3.25–4.5)	0.62	3.75 (3.0–4.5)	0.54	2.5 (1.67–3.5)	0.35	2.5 (1.75–3.5)	0.40	2.5 (2.0–3.5)	0.06	1.0 (1.0–1.5)	0.82
	single	4.0 (3.25–4.75)		4.25 (3.5–5.0)		3.63 (3.0–4.5)		4.0 (3.25–4.88)		2.0 (1.0–2.75)		2.13 (1.38–3.25)		2.0 (1.25–2.75)		1.13 (1.0–1.5)	
	widow/er	4.25 (3.5–4.75)		4.0 (3.5–5.0)		4.0 (4.0–4.5)		3.75 (3.25–4.0)		2.0 (1.75–3.33)		2.0 (1.5–2.5)		1.25 (1.0–4.75)		1.25 (1.0–1.75)	
Place of residence	median (25–75% IQR)																
	city	4.25 (3.75–5.0)	0.53	4.25 (3.5–4.75)	0.04	4.0 (3.5–4.5)	0.02	3.75 (3.25–4.5)	0.09	2.25 (1.5–3.25)	0.05	2.25 (1.5–3.25)	0.12	2.25 (1.5–3.0)	0.004	1.0 (1.0–1.5)	0.85
	village	4.0 (4.0–4.75)		3.38 (2.75–4.25)		3.25 (2.5–4.0)		3.13 (2.5–4.0)		2.63 (2.25–4.0)		3.0 (2.0–3.5)		3.13 (2.75–4.0)		1.0 (1.0–1.25)	
Cancer localization	median (25–75% IQR)																
	breast	4.0 (4.0–5.0)	0.86	4.5 (4.0–5.0)	0.12	4.38 (3.5–4.75)	0.31	3.88 (3.25–4.75)	0.44	1.75 (1.5–3.5)	0.17	1.75 (1.25–3.0)	0.01	2.0 (1.0–2.5)	0.03	1.0 (1.0–1.2)	0.10
	head and neck	4.0 (3.75–4.75)		3.88 (2.63–4.5)		4.0 (3.25–4.25)		3.88 (3.25–4.5)		2.75 (2.0–3.4)		2.5 (2.0–3.5)		2.55 (2.13–3.5)		1.25 (1.0–2.63)	
	lung	4.0 (3.75–5.0)		4.0 (3.5–4.75)		4.0 (3.0–4.5)		3.5 (2.75–4.75)		2.75 (2.0–4.0)		3.25 (2.0–3.5)		3.0 (1.5–3.25)		1.25 (1.0–1.75)	

IQR—interquartile range; r—correlation coefficient.

**Table 3 ijerph-19-09243-t003:** Coefficient of correlation of cognitive strategies among cancer patients.

		Acceptance	Positive Refocusing	Putting into Perspective	Refocus on Planning	Self-Blame	Rumination	Catastrophizing	Blaming Others
Acceptance	r	1							
	*p*-value								
Positive refocusing	r	0.415	1						
	*p*-value	<0.001							
Putting into perspective	r	0.506	0.383	1					
	*p*-value	<0.001	0.001						
Refocus on planning	r	0.075	0.407	0.190	1				
	*p*-value	0.519	<0.001	0.101					
Self-blame	r	0.040	−0.212	−0.123	−0.032	1			
	*p*-value	0.730	0.066	0.289	0.783				
Rumination	r	−0.015	−0.244	−0.112	0.151	0.406	1		
	*p*-value	0.895	0.033	0.334	0.193	<0.001			
Catastrophizing	r	0.017	−0.257	−0.132	0.038	0.333	0.759	1	
	*p*-value	0.881	0.025	0.255	0.747	0.003	<0.001		
Blaming others	r	0.009	0.156	0.025	0.269	0.125	0.311	0.309	1
	*p*-value	0.941	0.177	0.830	0.019	0.282	0.006	0.007	

r—correlation coefficient.

## Data Availability

Not applicable.

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
