# Peer review of "Identification of Cognitive Strategies Used by Cancer Patients as a Basis for Psychological Self-Support during Oncological Therapy"

_ijerph, 2022, doi:10.3390/ijerph19159243_

Round 1

Reviewer 1 Report

The article focused on identifying cognitive strategies among cancer pts during radiotherapy. The research methods were proper and adequately described. The results were clearly presented as box plots and Coefficient of correlations, and the statistical analyses were correct. The conclusion that cancer pts used adaptive cognitive strategy was supported by their results.

Interestingly, the authors observed differential copying strategies among different subgroups of pts observed. The discussion while citing multiple relevant studies on similar topics was comprehensive. 

I really enjoy reading this lovely research article. I applaud the authors for such good work. I recommend acceptance to the journal. It will attract broad interests among our readers.

Author Response

Thank you very much for reviewing our manuscript and for your kind comments.

Reviewer 2 Report

Dear authors, 

Thank you for giving me the opportunity to read this interesting paper. I have some queries :

1) How were the sample population conceptualized ? I am interested to know why cancer patients who underwent radical radiotherapy were selected.

2) In the results section ( Figure 2 ), there are far too many diagrams included, resulting in a lack of clarity. The authors are advised to summarize the results into fewer IQR diagrams for the sake of conciseness and brevity.

3) It is surprising that mental health distress markers such as anxiety and depression were not measured quantitatively in this study. The authors concluded that it is imperative to recognize coping strategies used by cancer patients in distress. Given that these variables ( anxiety/depression) were not directly measured in this study, this conclusion is incorrect and needs to be reworded.

Author Response

Thank you very much for your revision. Below please find our answers and explanations.

Ad 1. More than half of cancer patients are treated by radiotherapy. Radiotherapy most often is used in combination with other cancer treatment. Radiotherapy usually takes some time, patients during radiotherapy usually are in a good general condition and they are able to fulfill exhaustive questionnaire. Therefore we decided to conduct this study among cancer patients undergoing radiotherapy.

Ad 2. We believe that the data presentation in form of these diagrams is a little bit complicated whereas it gives a lot of information.

Ad 3. Anxiety and depression were not evaluated in our study. It is well known that cancer patients are exposed to distress. There are some components which could influence on distress and it is possible that cognitive strategies are among them.

Reviewer 3 Report

First of all, I would like to thank you for the opportunity to review this research. In spite of being a research which I consider of great interest for the scientific community, I consider that the following modifications should be made before its publication: 

I believe that the introduction is too brief and is not approached in a correct way; all the variables should be linked together and the study should be well grounded theoretically. Also, from lines 58-61, it should be eliminated since the use of the instruments does not correspond to this section. 

For the material and method, Cronbach's alpha should be added to study the degree of reliability of the instruments used. It should also be noted that the normality of the sample has been studied by means of the Kolmogorv-Smirnov test.

Regarding the results and discussion, these sections are well written. The discussion is very well founded and responds to the results obtained. 

I believe that a section entitled "limitations and future perspectives" should also be added, where the difficulties encountered in carrying out the research are discussed, as well as future research or perspectives that are carried out on the basis of the present study. 

Author Response

Thank you very much for your revision. Below please find our answers and explanations. The proper paragraphs in the manuscript were corrected due to suggestions.

The introduction was re-edited.

In material and method Cronbach alpha was added. It was noted that the normality of the sample has been studied by Shapiro-Wilk test (equal to the Kolmogorov-Smirnov test).

The section limitations and future perspectives was added. 

Round 2

Reviewer 2 Report

Dear authors,

Thanks for your revisions. However, my queries have not been satisfactorily answered.

1) Please summarize the figures to be more concise. I find the presentation of the results difficult to follow.

2) The authors seem to think that all cancer patients are naturally distressed, hence their assumption that the patients in their cohort utilized cognitive strategies to reduce their emotional distress. If the variables of depression/anxiety/psychological distress were not measured, these should be stated as an important limitation of the study.

Author Response

I would like to thank you for review and valuable comments.

Ad. 1. We added Table 2 - summarizing the data from the figures.

Ad 2. We agree that  the lack of measurement of variables depression/anxiety/psychological distress limited our study. We added that in Limitations and future perspectives section.

Reviewer 3 Report

Dear authors, 

First of all, I would like to congratulate you on your work. I consider that the article has been significantly improved and I think that it is now ready for publication. 

Best regards.

Author Response

I would like to thank you for review and valuable comments.